# Embryonic and post-embryonic development in the parasitic copepod *Ive ptychoderae* (Copepoda: Iviidae): Insights into its phylogenetic position

**Yu-Rong Cheng**[1]*, **Ching-Yi Lin**[2], **Jr-Kai Yu**[2,3]*

1 Department of Fisheries Production and Management, National Kaohsiung University of Science and Technology, Kaohsiung, Taiwan, 2 Institute of Cellular and Organismic Biology, Academia Sinica, Taipei, Taiwan, 3 Institute of Cellular and Organismic Biology, Marine Research Station, Academia Sinica, Yilan, Taiwan

* yrcheng@nkust.edu.tw (Y-RC); jkyu@gate.sinica.edu.tw (J-KY)

**Data Availability Statement:** All relevant data are within the paper.

**Funding:** JKY was supported by intramural funding from the Institute of Cellular and Organismic

## Abstract

Parasitic copepods are frequently discovered in many marine animals, and they exhibit great species diversity with remarkable morphological adaptations to their parasitic lifestyle. Similar to their free-living relatives, parasitic copepods usually develop through complex life cycle, but they eventually transform into a modified adult form with reduced appendages. Although the life cycle and distinct larval stages have been described in a few species of parasitic copepods, particularly those infecting commercially valuable marine animals (such as fishes, oysters, and lobsters), very little is known about the developmental process of the species that transformed into extremely simplified adult body plan. This paucity also causes some difficulties when investigating the taxonomy and phylogeny of this kind of parasitic copepods. Here we describe the embryonic development and a series of sequential larval stages of a parasitic copepod, *Ive ptychoderae*, which is a vermiform endoparasite living inside the hemichordate acorn worms. We devised laboratory regimes that enable us raising large quantity of embryos and free living larvae, and obtaining post-infested *I. ptychoderae* samples from the host tissues. Using defined morphological features, the embryonic development of *I. ptychoderae* can be categorized into eight stages (1-, 2-, 4-, 8-, 16- cell stages, blastula, gastrula, and limb bud stages) and the post-embryonic development comprises six larval stages (2 naupliar and 4 copepodid stages). Based on the comparisons of morphological characters in the nauplius stage, our results provide evidence to support that the *Ive*-group is more closely related to the Cyclopoida, which represents one of the two major clades that contain many highly transformed parasitic copepods. Thus, our results help to resolve the problematic phylogenetic position of the *Ive*-group in previous study based on analysis using 18S rDNA sequences. Combining with more molecular data, future comparative analyses on the morphological features of copepodid stages will further refine our understanding of the phylogenetic relationships of parasitic copepods.

Biology (https://icob.sinica.edu.tw/Eng) and the Career Development Award from Academia Sinica, Taiwan (98-CDA-L06)(https://www.sinica.edu.tw/en), and the grant from the National Science and Technology Council, Taiwan (101-2923-B-001-004-MY2)(https://www.nstc.gov.tw/?l=en). The funders play no role in the study design, data collection and analysis, decision to publish, or preparation of the manuscript.

**Competing interests:** The authors have declared that no competing interests exist.

## Introduction

Copepoda is one of the most diverse and abundant animal group in marine ecosystems and also the most common and widespread crustacean living in parasitic relationship with other organisms [1,2]. Parasitic copepods feed on host mucous, tissues, and blood, and their infection sometimes lead to mass mortalities resulting in change of host population [3–5]. Parasitic copepods exhibit various reductive tendencies, such as simplification and reduction of adult appendages [1,2,6]. This often poses a major obstacle to research on taxonomy and systematics of these animals, since classification of copepods is mainly based on the differences in morphology of adult females. It has been particularly challenging when dealing with the taxonomy and phylogeny of the two copepod orders, Poecilostomatoida and Cyclopoida, which include many species with highly transformed adults [7]. Traditionally, a cladistics phylogeny of ten copepod orders (Calanoida, Cyclopoida, Gelyelloida, Harpacticoida, Misophrioida, Monstrilloida, Mormonilloida, Platycopioida, Poecilostomatoida, and Siphonostomatoida) were proposed based on the morphological homologies in the body plan, segmentation and setation of appendages [6]. However, recent molecular analyses have shown some conflicting results with this phylogenetic scheme and questioned the validity of Poecilostomatoida [8,9]. Nowadays, the order Cyclopoida is generally considered encompassing Poecilostomatoid families, as suggested by several phylogenetic studies [10–13].

In those parasitic copepods with transformed body plan or with extremely reduced appendages, their larvae may provide rich information of additional morphological, behavioral, and ecological characters. The embryogenesis and post-embryonic stages (including nauplius and copepodid stages) of copepods have been investigated by many authors [7,14–23]. The morphological features of larval stages have been considered as useful characters for understanding the taxonomy and phylogeny [7,21,24–26]. Naupliar characters could represent plesiomorphic states and nauplii of closely related taxa are more likely to conserve synapomorphies than their adults [26]. To date, however, there are only a few parasitic copepods whose life cycle and larval stages have been thoroughly described (e.g. [16,27–31]). This shortage is mainly due to the difficulty of obtaining live ovigerous females and their eggs for life cycle research. Therefore, for those problematic parasitic copepods, studies on their life cycle and larval morphology would not only advance our knowledge of their life histories, but also may provide key information for resolving their phylogeny.

Previously study identified a new species of parasitic copepod, *Ive ptychoderae*, living inside the acorn worm *Ptychodera flava* (Hemichordata, Enteropneusta) collected from Penghu Islands in Taiwan [32]. *Ive ptychoderae* exhibits a highly transformed body plan (vermiform) which lacks apparent body segmentation and possesses reduced appendages since it lives in acorn worm. Characterized by having a reduced maxilliped and five pairs of annular swellings along the body, *I. ptychoderae* can be distinguished from the two previously reported parasitic copepods in acorn worms [32].Those two endo-parasites had been described more than one hundred years ago: *Ive balanoglossi* was collected from *Glossobalanus minutus* Kowalevsky, 1866 in the Mediterranean Sea [33]; *Ubius hilli* was described as a parasite of *Balanoglossus australiensis* Hill, 1894 in Australian waters [34]. Both of these two monotypic genera possess highly transformed body plan and reduced appendages, and their phylogenetic affiliations remain unsettled. Traditionally, the Pennellidae and Chondracathidae, had been used as convenient repositories for problematic parasitic copepods [35]. Due to the absence of clear external segmentation, *Ive* was originally grouped with Pennellidae by Mayer [33], but subsequently Kesteven [34] preferred to classify *Ive* and *Ubius* as members of the Chondracanthidae. However, due to the lack of definitive diagnostic features, eventually these two genera were not allocated to any family in the Copepoda and temporarily grouped together into

the *Ive*-group by later researchers [35]. In our recent study, we attempted to investigate the phylogenetic position of the newly discovered *Ive ptychoderae* by combining the results from morphological features of adults and molecular data [32]. Our molecular phylogenetic analysis showed that *I. ptychoderae* is likely a member of the Poecilostomatoida, but its 18S rDNA sequence is so distinct that it cannot be associated with any copepod family with known 18S rDNA sequence. Therefore, a new family, Iviidae, was proposed then to accommodate both *Ive* and *Ubius*. Nevertheless, until now the *Ive*-group is still treated as problematic taxa and its phylogenetic position remains ambiguous.

In this study, we seek to examine the life cycle of *I. ptychoderae* and utilize the morphological characters of its larvae to improve our understanding of the phylogenetic affiliation of this parasitic copepod. The first naupliar stage of *I. balanoglossi* and *U. hilli* were described previously by Mayer [33] and Kesteven [34], respectively. Other than that, nothing is known about the embryonic and larval development in the *Ive*-group. Here, we describe the life cycle of *I. ptychoderae* from embryonic stages, nauplius I to copepodid IV larval stages, as well as the morphological features of these developmental stages. In comparison with other copepods, we also discuss the phylogenetic implications of our results.

## Materials and methods

### Sample collection

Acorn worms *P. flava* were collected from the sandy beach at Chito, Penghu Islands, Taiwan (23°38'54.17"N; 119°36'14.40"E). To isolate the parasitic copepods, acorn worms were anesthetized with 0.2 M magnesium chloride in seawater for 15 minutes and the parasitic copepods were obtained from dissected cysts of hosts. The collection and use of invertebrate deuterostome animals for scientific research in JKY laboratory was reviewed by the Institutional Animal Care & Use Committee, Academia Sinica, and a waiver of ethic approval was granted (Case number: 16-04-956). The experimental procedures for handling the acorn worm *P. flava* were approved by the Institutional Biosafety Committee, Academia Sinica (Permit number: BSF0418-00003976).

### Incubation experiments

Twenty ovigerous females were obtained and cleaned off the host mucus in 3.5 cm Petri dishes filled with filtered/sterilized sea water. The egg sacs removed from the females were placed in the dishes and kept in water of 25±1 practical salinity unit in the room temperature (about 25°C). The water in the dishes was changed and unhatched eggs and exuviae were removed with a pipet every 12 hrs.

### Duration of embryonic and naupliar stages

For determining the duration of embryonic stages (1-, 2-, 4-, 8-cell, and so on), DAPI (4′, 6-diamidino-2-phenylindole) visualization of nuclei in fixed embryos were used to create reference stages of embryonic development. The embryonic development was monitored under dissection microscopy at 0.5–1 hour intervals and the duration time of embryonic development was documented. When the eggs developed into next stage of embryonic development, they were fixed in 4% Paraformaldehyde (PFA) for 12 hrs and then transferred into 96-well plates containing 100 µl of DAPI staining solution (5 µg/ml) at 4°C in the dark overnight. The de-ionized (DI) water was used as a staining vehicle (creating an osmotic imbalance) to stress the egg chorion and to allow fluorescent stain molecules into the eggs and nuclei. The stained eggs were washed by 1% phosphate-buffered saline with Tween® detergent (PBST) and

examined for stages under epi-fluorescence microscope. The higher resolution images were made under confocal microscope with UV excitation at 405 nm.

In the case of mature eggs, a total of 6,371 nauplii (first naupliar stage) were obtained during the first 70 hrs after the egg sacs were removed from the females. The nauplii were maintained in seawater at same conditions and treatments. The naupliar development was monitored under dissection microscopy every 8 hrs and the duration time was documented. When nauplii molted into infective stage (copepodid larvae), we reintroduced them to *P. flava* (re-infection study).

### Re-infection study

The devised re-infection regime allowed *I. ptychoderae* to complete its life cycle in the host tissues, enabling us to obtain post-infested copepodids for the morphological examination described below. In the re-infection study, ten individuals of *P. flava* were cut into 5 fragments (each about 1 cm) as surrogate hosts and infected by 10 copepodids, respectively. After infection, three infected acorn worms were sacrificed and examined every day. The acorn worms were placed in 3–5% ethanol solution for at least 4 hrs and washed with distilled water. Then, the washed water was poured through a fine net (mesh size = 100 μm) and examined under a dissection microscope. The copepods were picked up by forceps and preserved in 70% ethanol.

### Morphological studies

All preserved specimens (nauplii and copepodids) were kept in 85% lactic acid for at least 1 hour to soften the carapace before taking measurements and further dissection. The removed body parts and appendages were examined under a compound microscope [36]. All drawings were made with the aid of a drawing tube.

In this paper, the abbreviations NI and NII are used instead of nauplius I and II stages; the abbreviations CI to CIV are used instead of copepodid I to IV stages.

## Results

### Eggs and the duration of embryonic developments

The eggs of *I. ptychoderae* are usually laid in a pair of egg sacs carried by the female in the host. However, in some cases, the eggs are also shed free around the female body. Each egg sac contains approximately 600 eggs (see [32]) and the eggs are generally 128 (120–142) μm in diameter (*n* = 30, Fig 1).

The life history of *I. ptychoderae* was commenced from the time when the eggs were laid into the egg sacs. Based on our observation, the eggs in the same sac were non-synchronous in development. However, the nuclei and cleavages of eggs at all development stages were clearly visible, the embryonic development was easily separated into 8 stage categories: 1-, 2-, 4-, 8-, 16- cell stage, blastula, gastrula, and limb bud stages (Fig 1). The total duration of embryogenesis was 70 hrs from 1 cell stage to limb bud stage where limb bud stage accounted for more than half of the development time (37 hrs) (Fig 2).

### Morphology of nauplius stages, copepodid stages, and adult stage

The naupliar larvae hatched out from the mature eggs at the end of late limb bud stage. We made continuous observations of naupliar molting and identified two free-swimming naupliar stages and four copepodid stages (Figs 3–10). They are described separately as follows:

Nauplius I

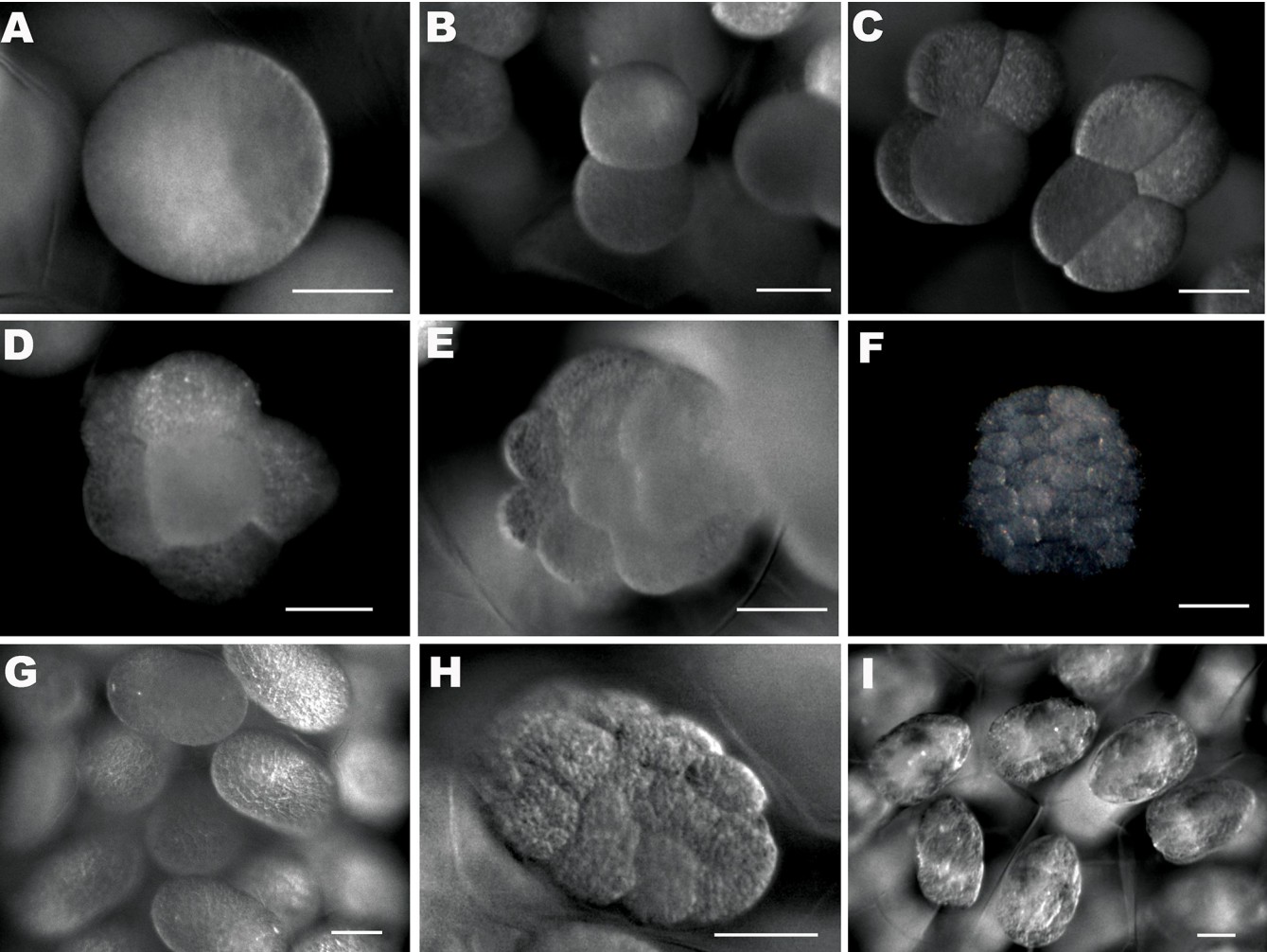

**Fig 1. The embryonic stages of Ive ptychoderae.** (A) One-cell stage; (B) 2-cell stage; (C) 4-cell stage; (D) 8-cell stage; (E) 16-cell stage; (F) Blastula stage; (G) Gastrula stage; (H) Early limb bud stage; (I) Late limb bud stage. Scale bars: A = 0.5 mm; B–I = 0.05 mm.

Fig 4.

Body (Fig 4A) oval or pear-shaped, full of yolky granules (see Fig 3B), longer than wide, Average length 163 (161–166) μm and greatest width 105 (103–107) μm based on 10 specimens. Three pairs of swimming appendages: antennule, antenna, and mandible. Labrum (Fig 4A) curved prominence without ornamentation. Caudal armature (Fig 4A) represented by 1 pair of fine setae. Antennule (Fig 4A and 4B) uniramous, 1-segmented, armed with 1 proximoventral, 1 midventral and 2 unequal terminal setae. Antenna (Fig 4A and 4C) biramous, with exodopod slightly longer than endopod; coxa without masticatory process, separate from short basis; exopod 4-segmented, first segment elongated and fused with basis, each segment armed with 1 seta; endopod 2-segmented, first segment unarmed, indistinctly separate from basis, second segment armed with 2 plumose setae. Mandible (Fig 4A and 4D) biramous, coxa separate from basis, both coxa and basis unarmed. Exopod 5-segmented, each armed with 1 plumose seta except for first segment; first segment elongated, fused with basis. Endopod 1-segmented, armed with 1 proximoventral, 1 midventral, and 2 equal terminal setae.

Nauplius II

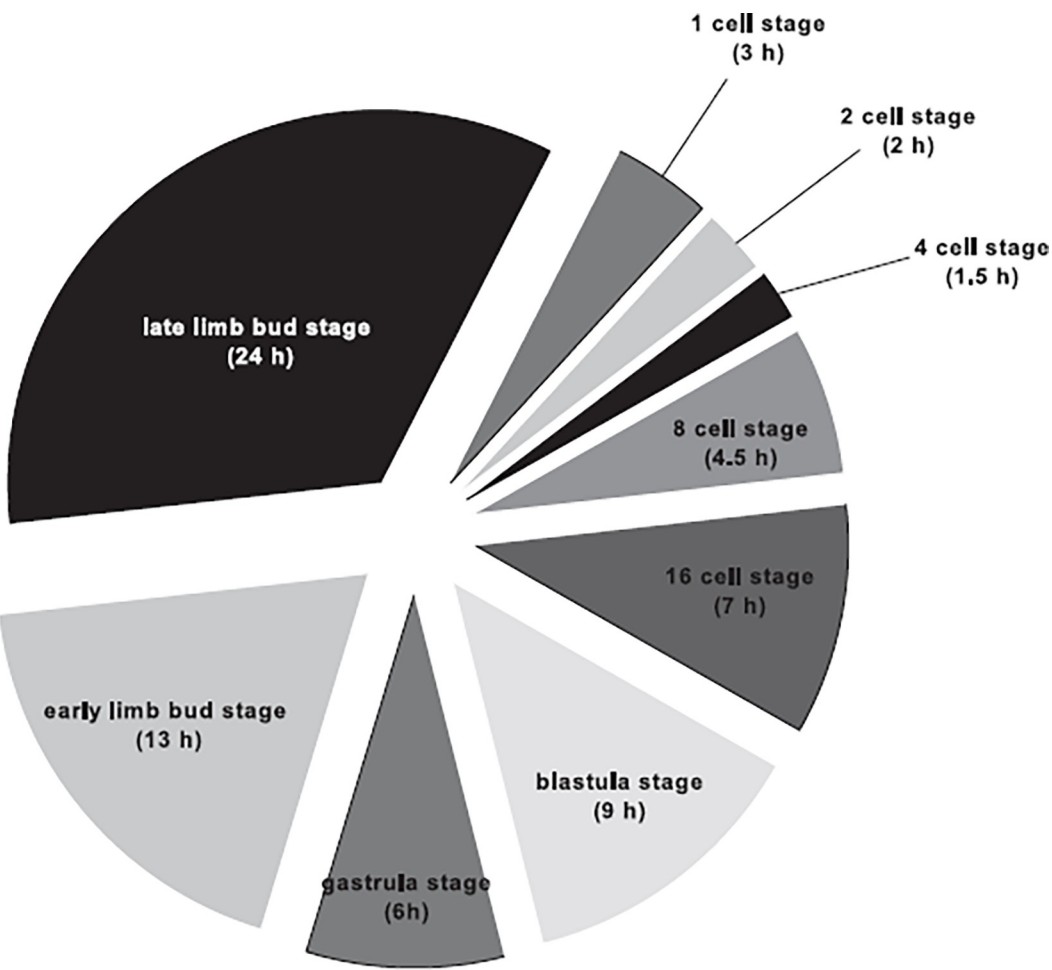

**Fig 2. The duration of embryonic stages (1-, 2-, 4-, 8-, 16- cell, blastula, gastrula, and limb bud stages) of *Ive ptychoderae*.** The assigned time of each stage is indicated in parentheses.

Fig 5.

Body (Fig 5A) as in NI but slightly slender, average length 160 (158–163) μm and greatest width 94 (92–97) μm based on 10 specimens. Labrum and caudal armature (Fig 5A) as in NI. Antennule (Fig 5A and 5B) 2-segmented, first segment with 2 rows of spinules and 3 equal setae; second segment with 3 rows of spinules, one sub-distal seta (largest), and 2 unequal terminal setae. Antenna (Fig 5A and 5C) as in NI, without ornamentation on coxa and basis; exopod distinctly 5-segmented, with 1 plumose seta at inner tip of segments 1–4 and 2 unequal smooth seta on fifth segment; endopod 2-segmented, first segment unarmed, second segment armed with 1 middle smooth seta and 2 terminal plumose setae. Mandible (Fig 5A and 5D) as in NI.

Copepodid stages

Definition of the copepodid stages was based on the examination of morphological characters of antennule, antenna, legs 1 and 3 under the light microscopy. From our own observations, we determined that the copepodid stages consist of 4 stages. They are described separately as follows:

Copepodid I

Fig 6.

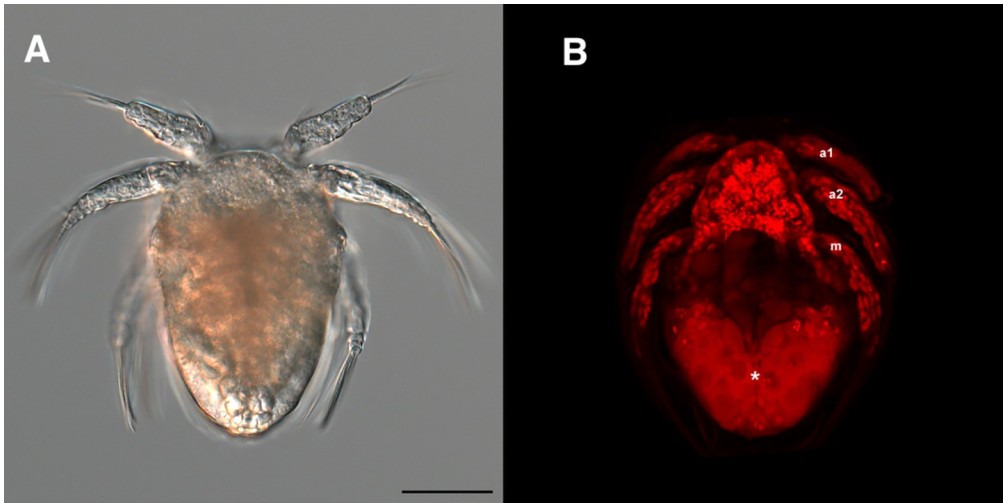

**Fig 3.** Images of Nauplius I stage larva of *Ive ptychoderae*; (A) Bright-field image, and (B) fluorescent image by confocal microscopy with nucleus staining (red). (a1) Antennule; (a2) Antenna; (m) Mandible; (*) Yolky granules. Scale bar = 50 μm.

Body (Fig 6A) with broad flattened prosome. Average length of 10 specimens measured 260 (248–268) μm (excluding seta) and greatest width 93 (91–95) μm. Prosome 2-segmented. Cephalothorax distinctly longer than wide, 130×93 μm; second prosomal somite wider than long, 68×33 μm. Urosome (Fig 6A and 6B) 3-segmented, from anterior to posterior 85 μm. Third segment longest, about 38 μm in length and 24 μm in width. Caudal ramus (Fig 6A–6C) small, about 15 × 11 μm, bearing with 6 setae, innermost terminal seta longest, consisting of thicker proximal part and slender distal flagellum. Surface of body unornamented (Fig 6A).

Antennule (Fig 6D) 4-segmented, with setal formula: 6, 4, 2+1 aesthetasc, and 6+1 aesthetasc, all setae naked. Antenna (Fig 6E) 4-segmented, consisting of coxobasis and 3 endopodal segments, each endopodal segments with 1 claw and 1 seta; exopod vestigial and unarmed.

Labrum (Fig 6F) with concave posterior margin and well developed rounded lateral lobes. Mandible, maxillule, and maxilliped absent.

Legs 1 and 2 (Fig 6G–6I) biramous, each ramus with 1-segmented exopods and endopods. Formula of spines (in Roman numerals) and setae (in Arabic numerals) as follows:

|       | Coxa | Basis | Exopod | Endopod |
|-------|------|-------|--------|---------|
| Leg 1 | 0–0  | 1–0   | IV-3   | III-4   |
| Leg 2 | 0–0  | 1–0   | IV-3   | III-3   |

Legs 3 and 4 absent. Leg 5 (Fig 6A) reduced to minute process bearing a terminal seta and an adjacent dorsal seta.

Copepodid II

Fig 7.

Body plan (Fig 7A and 7B) as in CI, average length 244 (228–260) μm and greatest width 91 (89–93) μm based on 10 specimens. Caudal ramus (Fig 7B) as in CI.

Antennule (Fig 7C) armature and segmentation as in the previous stage, except for an aesthetasc instead of seta on second segment. Antenna (Fig 7D) 3-segmented, consisting of coxobasis and 2 endopodal segments; endopod first segment unarmed, second segment divisible into 3 parts: proximal part carrying 1 small claw; middle part armed with 1 seta and 1 claw;

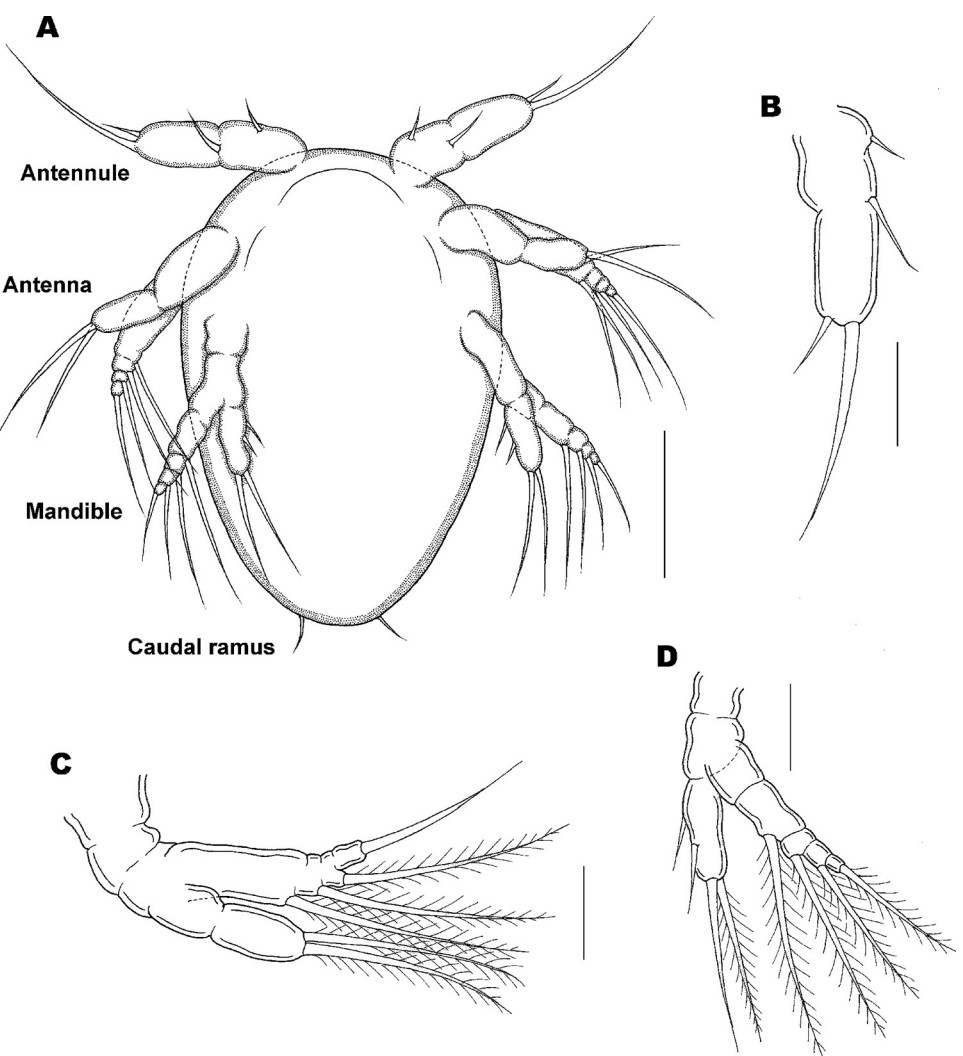

**Fig 4. Nauplius stage I of *Ive ptychoderae*.** (A) Habitus, ventral; (B) Antennule; (C) Antenna; (D) Mandible. Scale bars: A = 0.05 mm; B–D = 0.025 mm.

and distal part with 1 slender claw and 1 longest seta. Labrum as in CI. Mandible, maxillule, and maxilliped not present. Legs 1–5 as in the previous stage.

Copepodid III

Fig 8.

Body (Fig 8A) moderately slender. Average length 218 (206–228) μm and greatest width 73 (70–75) μm based on 9 specimens. Segmentation of body indistinct. Caudal ramus (Fig 8A and 8B) elongated, bearing 1 terminal longest seta (= innermost terminal seta in the previous stages), and 2 inner, marginal setae in addition to 2 setae on outer-lateral margin.

Antennule (Fig 8C and 8D) 4-segmented, setal formula: 6, 4, 3, and 3+1 aesthetasc. Antenna (Fig 8C and 8E) 2-segmented, basal segment largest, bearing patch of spinules on middle margin, second segment with 1 long seta with lamella, 2 small spiniform setae, and 2 claws.

Labrum (Fig 8C) projected posteriorly at both free corners. Mandible, maxillule, and maxilliped not present.

Legs 1 and 2 (Fig 8F and 8G) biramous, with 1-segmented exopod and endopod. Formula of spines (in Roman numerals) and setae (in Arabic numerals) as follows:

|  | Coxa | Basis | Exopod | Endopod |
|---|---|---|---|---|
| Leg 1 | 0–0 | 0–0 | II-3 | 2 |
| Leg 2 | 0–0 | 0–0 | II-3 | 3 |

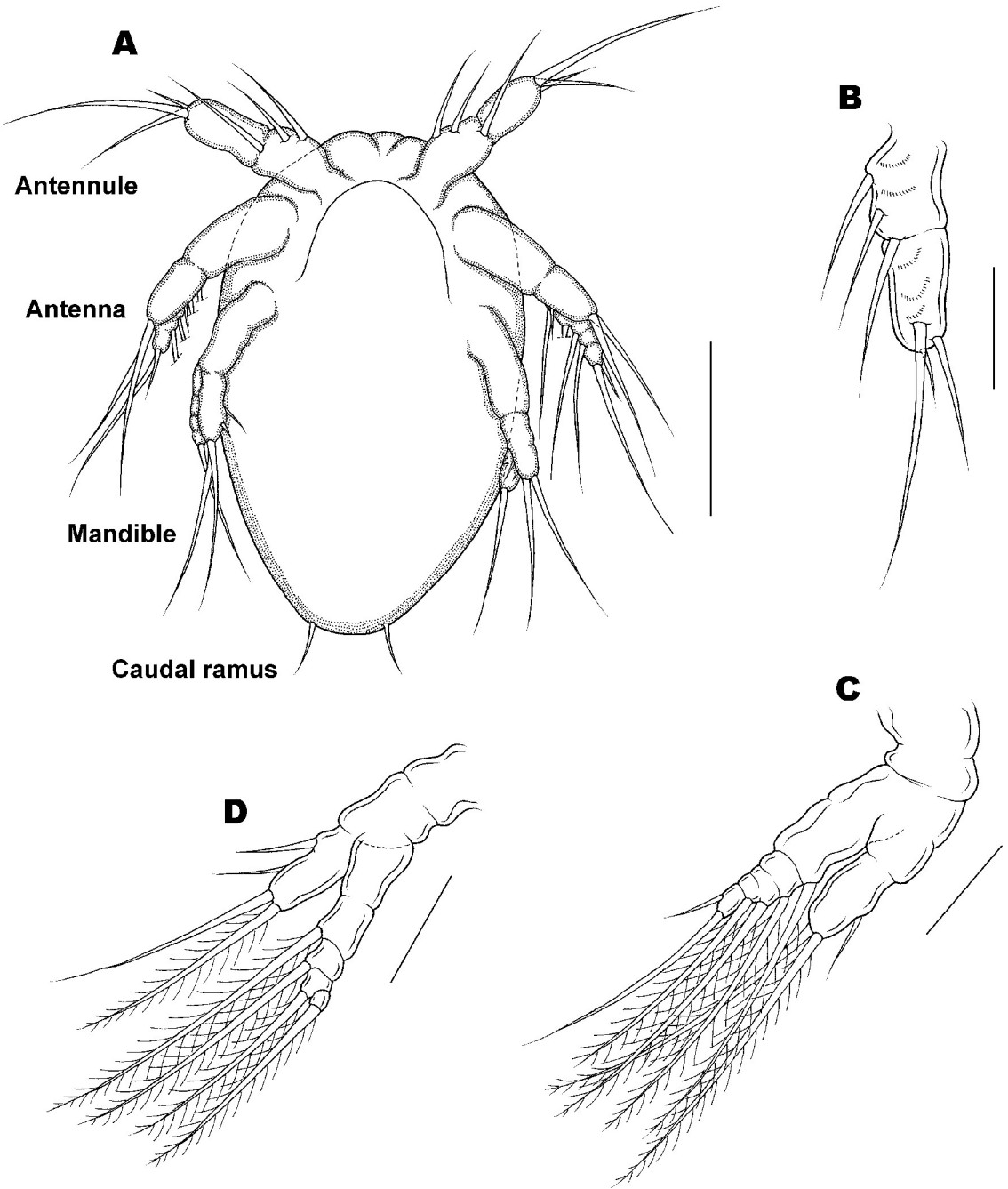

**Fig 5. Nauplius stage II of *Ive ptychoderae*.** (A) Habitus, ventral; (B) Antennule; (C) Antenna; (D) Mandible. Scale bars: A = 0.05 mm; B–D = 0.025 mm.

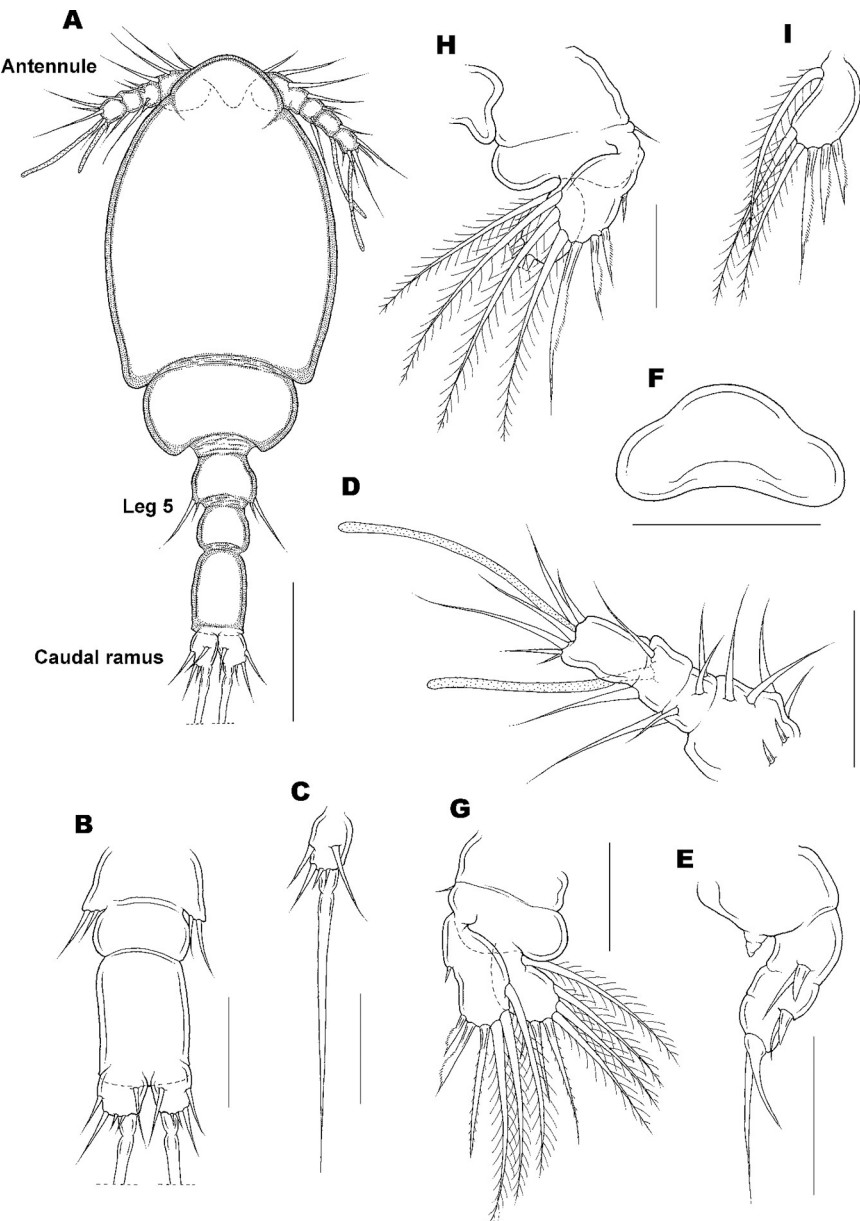

**Fig 6. Copepodid stage I of *Ive ptychoderae*.** (A) Habitus, dorsal; (B) Urosome, dorsal; (C) Caudal ramus; (D) Antennule; (E) Antenna; (F) Labrum; (G) Leg 1; (H) Leg 2; (I) Endopod of leg 2. Scale bars: A = 0.05 mm; B–I = 0.025 mm.

Legs 3 and 4 absent. Leg 5 (Fig 8A) reduced to minute process bearing a terminal seta and an adjacent small seta.

Copepodid IV

Fig 9.

Body (Fig 9A) slender than CIII. Average length 856 (465–1175) μm and greatest width 164 (107–190) μm based on 10 specimens. Segmentation of body indistinct. Caudal ramus (Fig 9B) carrying 1 inner, 2 lateral, and 2 terminal tubercles.

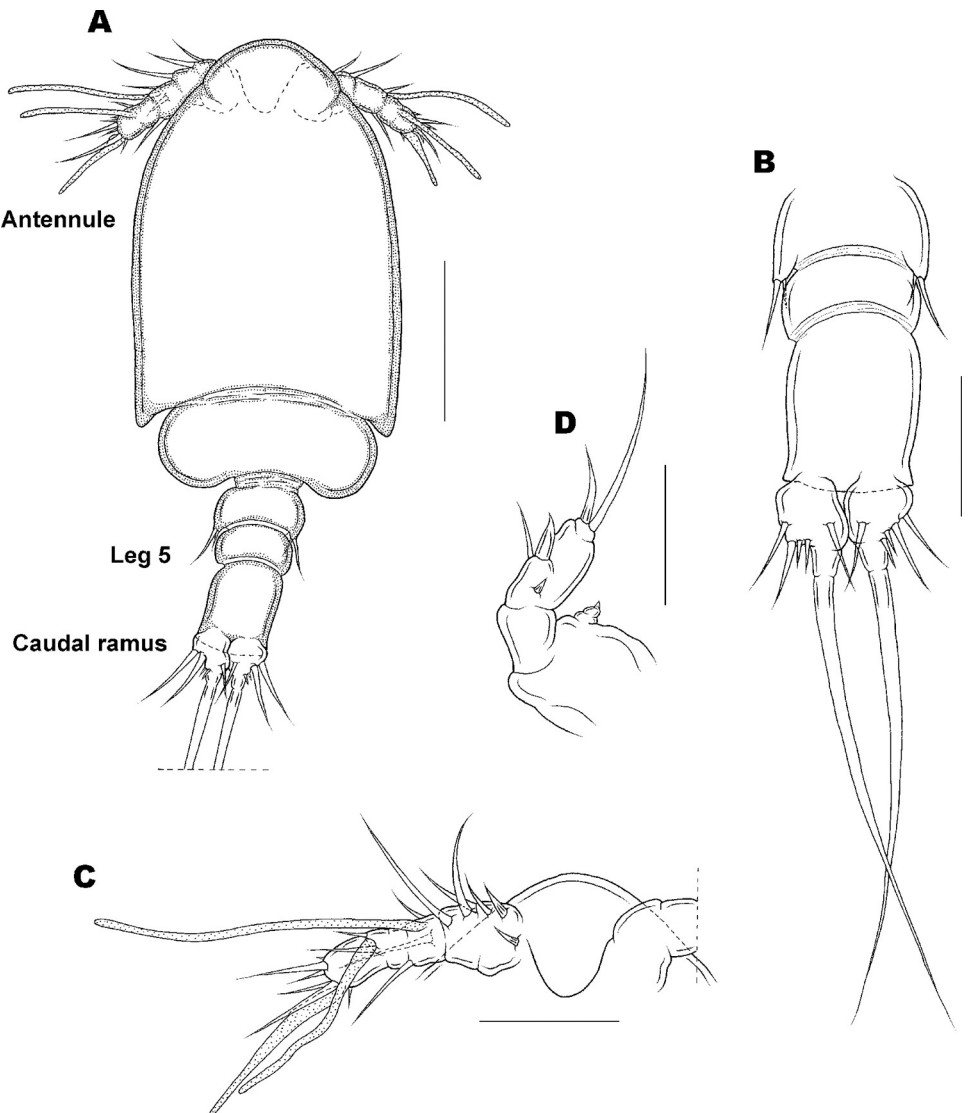

**Fig 7. Copepodid stage II of *Ive ptychoderae*.** (A) Habitus, dorsal; (B) Urosome, dorsal; (C) Antennule; (D) Antenna; Scale bars: A = 0.05 mm; B–D = 0.025 mm.

Antennule (Fig 9C) short, unsegmented, divisible into 3 parts: proximal part carrying 2 unequal setae; middle part armed with 1 large and 2 small setae; distal region with 3 small tubercles. Antenna (Fig 9D) 3-segmented, first and second segments unarmed; third segment with 1 large claw and 1 pointed process.

Labrum (Fig 9E) projected posteriorly at both free corners. Mandible, maxillule, and maxilliped not present. Legs 1 and 2 as in adult (Fig 9F and 9G; see [32]). Legs 3–5 absent.

Adult

The adult *I. ptychoderae* is vermiform, without external segmentation, lacking typical morphological features (such as mandible, maxillule, maxilla, maxilliped, and legs 3–5), resulting in simplification of body plan. The female is characterized by having five pairs of annular swellings along the body that are morphologically similar but distinguishable from its congeners in acorn worms [32]. More detailed descriptions of the adult morphology of *I. ptychoderae* were reported previously [32], and thus will not be repeated in the paper.

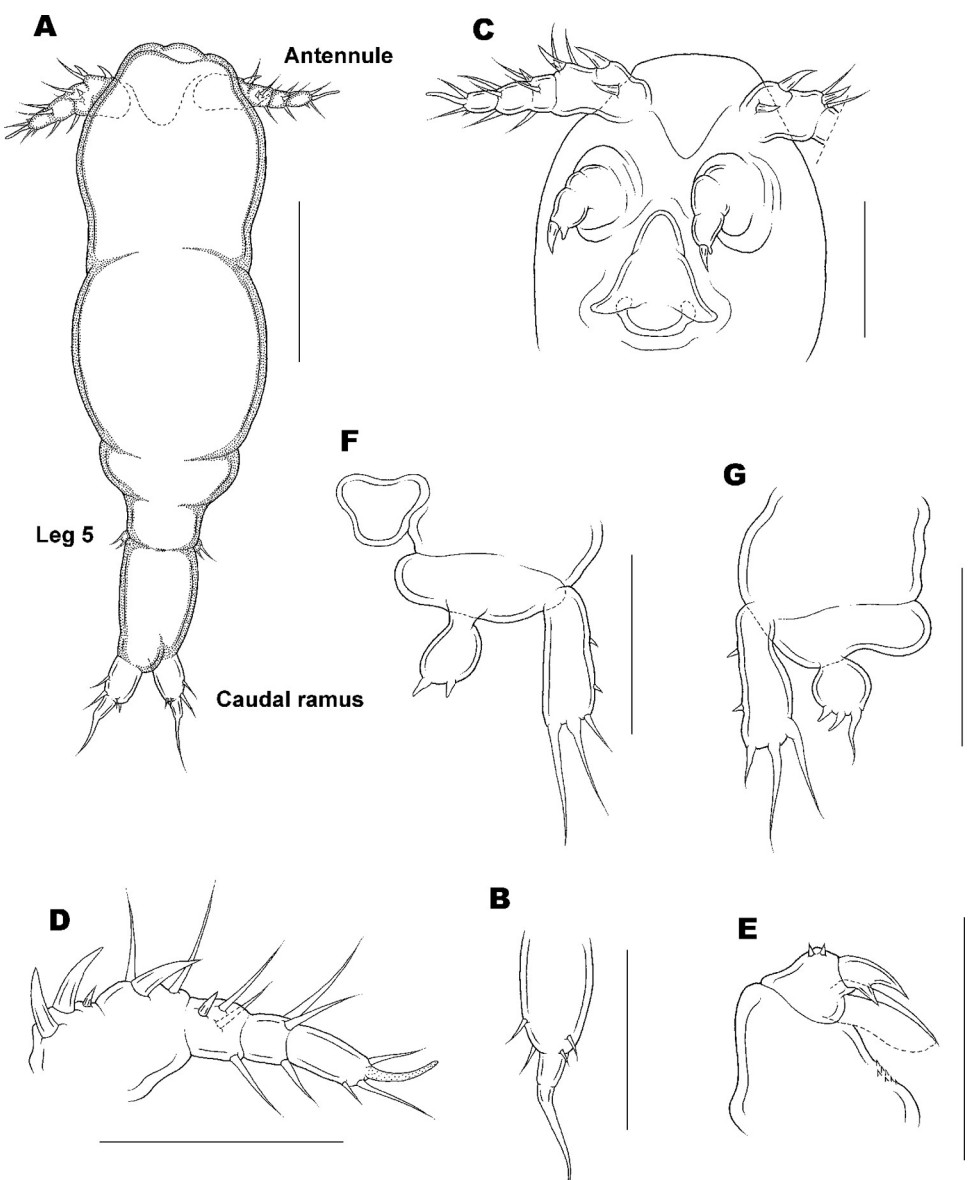

**Fig 8. Copepodid stage III of *Ive ptychoderae*.** (A) Habitus, dorsal; (B) Caudal ramus; (C) Antennal area and oral region, ventral; (D) Antennule; (E) Antenna; (F) Leg 1; (G) Leg 2. Scale bars: A = 0.05 mm; B–G = 0.025 mm.

## The duration of post-embryonic developments

In total, the duration of post-embryonic development lasted 38 d and included 6 stages, namely, NI and NII (free-swimming stage) and CI through CIV (infective stage) (Fig 10). The nauplii passed through 2 stages to become copepodid and underwent a single molt at every stage. The larvae of NI swam actively and were positively phototactic. However, the swimming activity decreased as the nauplii started to molt. Approximately 2 d, the NI molted into NII. The larvae of NII took an average 7 d for growing into the first infective stage (CI). The newly molted CI larvae were rather restless, wandering over the Petri dishes to search for a suitable microhabitat of host fragment by synchronously moving two pairs of legs. The duration of this stage to the next stage (CII) was relatively short (approximately 2 d) due to a few

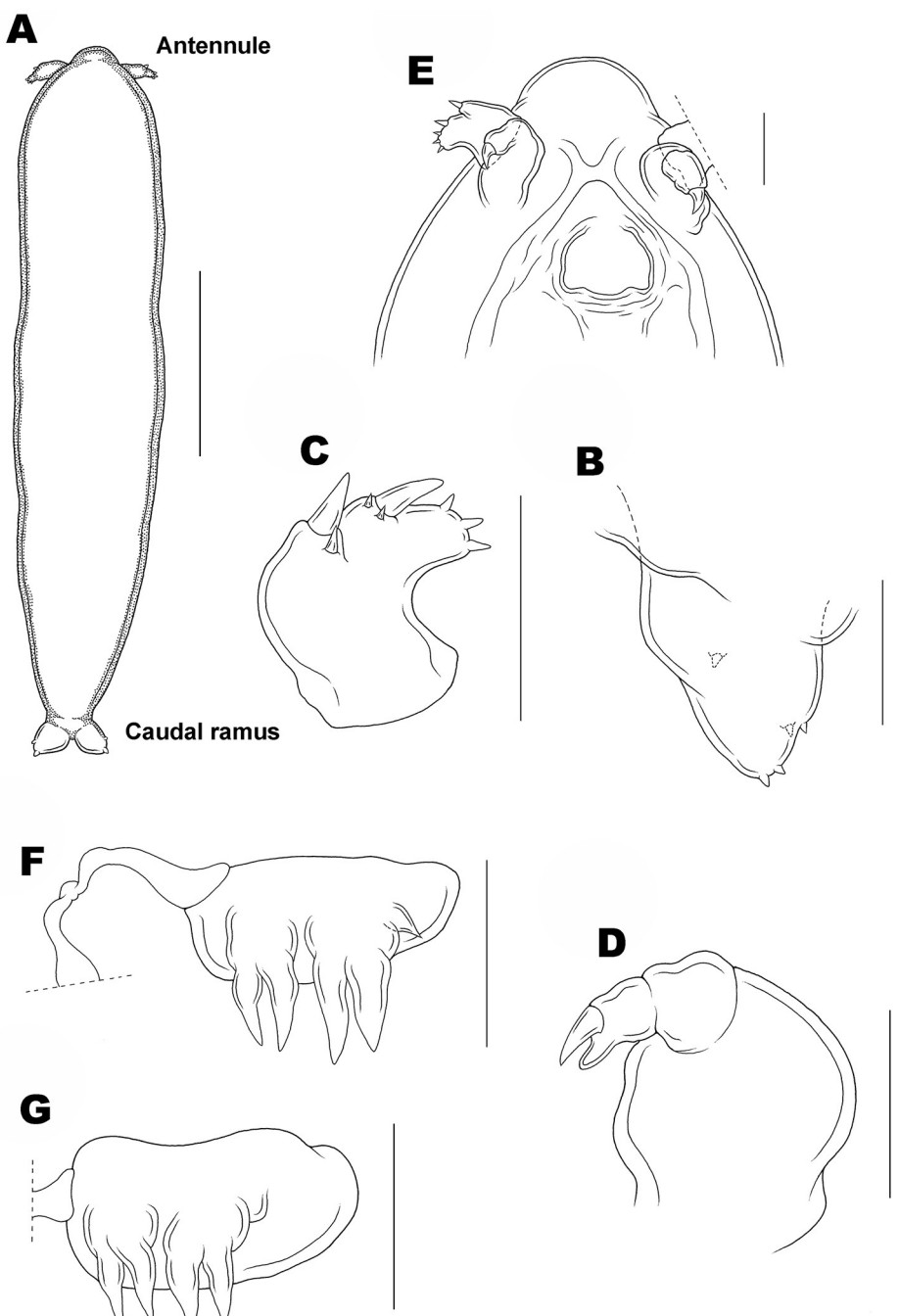

**Fig 9. Copepodid stage IV of *Ive ptychoderae*.** (A) Habitus, dorsal; (B) Caudal ramus; (C) Antennule; (D) Antenna; (E) Antennal area and oral region, ventral; (F) Leg 1; (G) Leg 2. Scale bars: A = 0.1 mm; B–G = 0.025 mm.

morphological changes. The differences between CI and CII only occurred in the length/width ratio of prosome and armature of antennule and antenna. However, there were dramatic differences in morphology and behavior since CII stage. The duration of CII to CIII (7 d) and CIII to CIV (9 d) were comparatively longer than that of CI to CII (2 d). Consequently, it took about 2 weeks in total from CII to CIV. It should be mentioned that the duration of CIV lasted

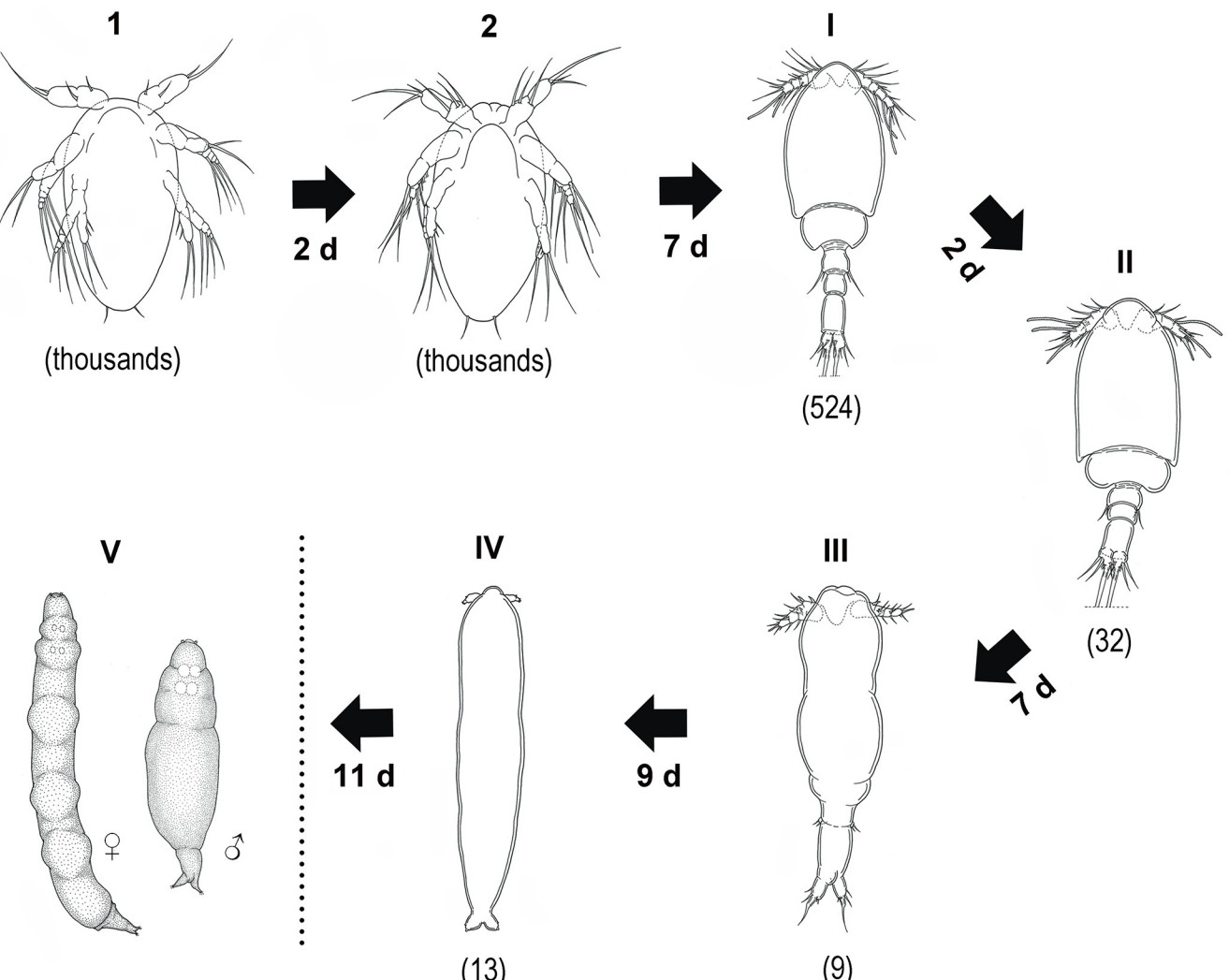

**Fig 10. Duration of post embryonic stages of *Ive ptychoderae*.** (1–2) naupliar stages; (I-IV) copepodid stages; (V) adult stage. The duration time (days) of each stage is provided and the number of copepod individuals at each developmental stage is indicated in parentheses. Dotted line indicates the end of incubator experiment.

at least for 11 d until the end of our experiment. The structures of appendages are highly similar to adult (see [32]), but the body size is much small, ranging from 0.47 to 1.76 mm.

## Discussion

Copepods represent a diverse animal group with great morphological variations and versatile life cycle strategies [7,37]. In this study, we described the life cycle of a parasitic copepod *I. ptychoderae* and the morphological features characterizing each developmental stage of this species. These new results provide important information to suggest the phylogenetic affiliation of this parasitic copepod.

### *Ive ptychoderae* exhibits a typical process of embryogenesis

It usually takes copepods about 24–72 hrs from eggs to hatching [38–40]. Buttino et al. [41] developed a new staining protocol for the visualization of copepod embryos with the confocal

laser scanning microscope. Since then, fluorescence probes have been used on the studies of copepod embryonic development [41,42]. However, this approach has rarely been used in the parasitic copepods. Here, we show that the embryos of parasitic copepod can also be rapidly assessed using nucleus staining under fluorescent microscope (Fig 3). We estimated that the pre-hatching embryonic development of *I. ptychoderae* is approximately 70 hrs, which is in line with previous observations in copepod species. And we observed that the limb bud stages accounted for approximately half of the embryogenesis period (about 37 hrs). This is conceivable because organogenesis occurs in limb bud stages and takes more time to develop complex structures of the larva, so that the newly hatched nauplius can be equipped with necessary organ systems for the subsequent planktonic phase of its life cycle [43].

### *Ive ptychoderae* reduces its naupliar and copepodid stages

The basic life cycle of symbiotic copepods encompasses two planktonic phases, the naupliar and copepodid stages, during which the animals may perform diverse behaviors including dispersal, infection, host switching, mating, and presumably, predator-avoidance [44–46]. In general, it has been known that symbiotic copepods typically have 12 post-embryonic stages including 6 nauplius, 5 copepodid stages, and then molt into adult stage [16–18,20–22,26,37]. However, abbreviated life cycle is characteristic for some parasitic copepods, thus they are highly efficient in locating and infecting new hosts without needing long-distance larval dispersal [7,17,21,22,28,29,47]. We also observed the reduction of naupliar and copepodid stages in *I. ptychoderae*. Based on the morphological characters described in this study, the post-embryonic development of *I. ptychoderae* can be categorized into 6 (2 nauplius and 4 copepodid) stages from NI to CIV stage. Becasue adult *I. ptychoderae* is an endo-parasite with highly transformed body plan and reduced appendages, the exhibition of a reductive life cycle is not unexpected. It may reflect its adaptation to the parasitism.

It has been noted that environmental factors, such as temperature, salinity, food type, and host availability, can affect pre- and post- embryonic development in copepods [48–50]. In this study, we only employed one standardized environmental regime to culture *I. ptychoderae* embryos and larvae for the observation. It would be interesting to manipulate the culturing condition in future research, to investigate whether environmental factors can also exert influence over *I. ptychoderae* development.

### The morphological characters of *I. ptychoderae* nauplius larvae and their implications to the phylogenetic affiliation of this highly transformed parasitic copepod

Izawa [7] suggested that two types of naupliar forms were defined based on the mode of nutritional supply or egg size (the amount of yolk granules). The copepod eggs less than 120 μm may yield feeding nauplii (planktotrophic form) which usually have the basic six naupliar stages. On the other hand, the copepod eggs with large size may yield lecithotrophic nauplii (non-feeding form) and reduce in the number of naupliar stages. Our results showed that the eggs of *I. ptychoderae* were larger than 120 μm, and that ample yolk granules were observed inside the nauplius stage larvae (Fig 3). This suggests that (1) the nauplii of *I. ptychoderae* are lecithotrophic form and do not feed during the nauplius stages, and (2) its naupliar stage may be less than six stages. Lecithotrophic nauplii are common to most parasitic copepods belonging to the orders Cyclopoida and Siphonostomatoida [7,19,29,35,51]. The body shape of nauplii of *I. ptychoderae* are fundamentally oval or pear-shaped, which is also considered as a typical morphology in the siphonostomatoids and cyclopoids [7,18,28]. However, the fish-parasitizing siphonostomatoids have evolved a remarkable trait by converting the copepodid

stages into a series of more effectively attached chalimus stages in their life cycles. Without recognizable chalimus stage, *I. ptychoderae* is not likely belonging to the order Siphonostomatoida and thus may be more closely related to the Cyclopoida.

Mayer [33] and Kesteven [34] described the first nauplius stage of *I. balanoglossi* and *U. hilli*, respectively. They provided a figure to show a whole animal in ventral view and the morphological information of appendages. Combining with the results from previous studies and our observations, the nauplii of *Ive*-group possess some common morphological features. For example, their bodies were full of yolky granules, lacked a masticatory process on the antennal coxa, and the labrums were reduced in size. However, the antenna of *Ive*-group with 2-segmented endopod in the nauplius stage (as in the Facetotecta and Cephalocarida) is a primitive character for distinguishing them from the existing orders of Copepoda. In addition, the exopod of antenna of *I. ptychoderae* is 4 segments in the first nauplius stage, but the generalized exopod is 5–6 segments in most copepod such as Siphonostomatoida, Cyclopoida, Calanoida, and Harpacticoida. Nevertheless, it should be noted that the exopod of antenna may decrease in the typical number of segments in the nauplii of some parasitic copepods [28,30,52]. The other characters in the naupliar stage of *Ive*-group also provided some signs to distinguish them from other copepod orders. According to Izawa [7], two type of apical setation of the mandibular exopod (with 1 or 2 setae) are recognized in the nauplii of copepods. By using this feature, systematic positions of some problematic copepods are resolved. For example, the Antheacheridae, Mesoglicolidae, and Herpyllobiidae have been correctly accommodated in the Cyclopoida (= Poecilostomatoida) based on their nauplii with one seta type; the Sponginticolidae (treated as order uncertain) has been removed to Siphonostomatoida since it exhibited the two setae type in the nauplius stage. Thus, the presence of one apical seta on the mandibular exopod in the nauplius stage of *Ive*-group is a main character for distinguishing them from the species belonging to the Calanoida, Harpacticoida, and Siphonostomatoida (with 2 setae). The number of apical elements of the antennule in the first nauplius stage of *Ive*-group with only 2 setae (instead of 3 or 4 setae) is similar to that of ergasilids in the Cyclopoida. Indeed, *Ive*-group might be more closely related to *Antelichomolgus*-group (including Clausidiidae, Oncaeidae, Taeniacanthidae, Tegobomolochidae, and Bomolochidae) belonging to the Cyclopoida based on four combinations of naupliar characters [7]. However, species in the *Antelichomolgus*-group have 6 naupliar stages, while there are only 2 naupliar stages in *I. ptychoderae*. Moreover, the nauplii of *I. ptychoderae* with 1-segmented mandibular endopod and 5-segmented mandibular exopod are significantly different from the general form in the cyclopoids (with 2-segmented mandibular endopod and 4-segmented mandibular exopod). All in all, based on the morphological characters of naupliar stages, *I. ptychoderae* exhibited many unique characters.

Dahms [26] pointed out several reasons to explain why naupliar characters have thus far been widely neglected in the systematic and phylogenetic studies: (1) It is difficult to obtain the detailed information on naupliar characters, (2) the nauplii may provide fewer characters than later ontogenetic stages, (3) It may lack appropriate comparative data, (4) non-feeding nauplii may lack phylogenetically valuable structures, and (5) conflicting evidence would occurred in comparing the character states between adult and nauplius stages. In the case of *I. ptychoderae*, the reason 3 is the most critical factor to limit the resolution of our results. The life cycles of crustacean parasite are largely unknown (1% of species) [22]. Taking sea lice for example, there are more than 450 caligid species but the complete life cycles are now known only for 17 species (3.8%) representing just three genera, *Caligus* Müller 1785 (12 species), *Lepeophtheirus* von Nordmann 1832 (four species) and *Pseudocaligus* Scott 1901 (= *Caligus*; one species) [17,53–55].

## Conclusion

In this study, we described the embryonic development and the entire series of larval stages of the parasitic copepod, *Ive ptychoderae*. Using morphological characters in the nauplius stage, we provide evidence to support that the *Ive*-group is more closely related to the Cyclopoida group. However, the findings in this study also demonstrate that the nauplii of *I. ptychoderae* indeed exhibit many distinguishable characters from the major existing orders of Copepoda. For a definitive view of the phylogenetic relationships of *Ive*-group, further phylogenetic discussion cannot be made on the basis of the known nauplii only. More information on the nauplii of other related copepods is indispensable for addressing this issue, and the features on the copepodid stages must also be taken into consideration. In addition, future molecular phylogenetic analysis including related species, such as *I. balanoglossi* and *U. hilli*, would be highly welcomed.

## Acknowledgments

We would like to thank Che-Huang Tung and Sharon Horng for their assistance in the field.

## Author Contributions

**Conceptualization:** Yu-Rong Cheng, Jr-Kai Yu.

**Data curation:** Yu-Rong Cheng.

**Formal analysis:** Yu-Rong Cheng, Ching-Yi Lin, Jr-Kai Yu.

**Funding acquisition:** Jr-Kai Yu.

**Investigation:** Yu-Rong Cheng, Ching-Yi Lin, Jr-Kai Yu.

**Methodology:** Yu-Rong Cheng.

**Project administration:** Jr-Kai Yu.

**Validation:** Yu-Rong Cheng.

**Visualization:** Yu-Rong Cheng, Ching-Yi Lin, Jr-Kai Yu.

**Writing – original draft:** Yu-Rong Cheng, Jr-Kai Yu.

**Writing – review & editing:** Yu-Rong Cheng, Ching-Yi Lin, Jr-Kai Yu.

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
