## [Decision Letter · Decision Letter 0]

4 Dec 2022

PONE-D-22-29055Embryonic and post-embryonic development in the parasitic copepod Iveptychoderae (Copepoda: Iviidae): insights into its phylogenetic positionPLOS ONE

Dear Dr. Yu,

Thank you for submitting your manuscript to PLOS ONE. After careful consideration, we feel that it has merit but does not fully meet PLOS ONE’s publication criteria as it currently stands. Therefore, we invite you to submit a revised version of the manuscript that addresses the points raised during the review process.

Thanks for considering Plose One for dissemination of your findings. The developmental stages of this family, Iviidae, and the life cycle of I. ptychoderae based on the morphological characters of larvae will fill the knowledge gap and help in the establishment of phylogenetic affiliation . The study is well designed and systematically done. These results would help to resolve the problematic phylogenetic position of the Ive-group, and showcase the approach of utilizing developmental characters (i.e.,

morphology of immature stages.

A sentence in the introduction section regarding the role of parasitic copepods on ecosystem structuring need tp be explained. Justify the the re-infection study in this Ms.

Besides the Manuscript will benefit from following published information: doi:10.1093/plankt/20.2.271 and Kumar (2003), doi:10.1127/0003-9136/2003/0157-0351.

I am personally curious to learn the post embryonic developmental rate as a function of (i) host availability (ii) temperature and salinity.

Overall the manuscript is well written , methods are easily reproducible .Please submit your revised manuscript by Jan 18 2023 11:59PM. If you will need more time than this to complete your revisions, please reply to this message or contact the journal office at plosone@plos.org. Please include the following items when submitting your revised manuscript:A rebuttal letter that responds to each point raised by the academic editor and reviewer(s). You should upload this letter as a separate file labeled 'Response to Reviewers'.A marked-up copy of your manuscript that highlights changes made to the original version. You should upload this as a separate file labeled 'Revised Manuscript with Track Changes'.An unmarked version of your revised paper without tracked changes. You should upload this as a separate file labeled 'Manuscript'.If applicable, we recommend that you deposit your laboratory protocols in protocols.io to enhance the reproducibility of your results. Protocols.io assigns your protocol its own identifier (DOI) so that it can be cited independently in the future. For instructions see: https://journals.plos.org/plosone/s/submission-guidelines#loc-laboratory-protocols. Additionally, PLOS ONE offers an option for publishing peer-reviewed Lab Protocol articles, which describe protocols hosted on protocols.io. Read more information on sharing protocols at https://plos.org/protocols?utm_medium=editorial-email&utm_source=authorletters&utm_campaign=protocols.

We look forward to receiving your revised manuscript.

Kind regards,

Ram Kumar, Ph.D. D. Sc (H/C)

Academic Editor

PLOS ONE

Journal Requirements:

"NO authors have competing interests". 

Overall the manuscript is well written , methods are easily reproducible .

Reviewers' comments:

Reviewer's Responses to Questions

**Comments to the Author**

1. Is the manuscript technically sound, and do the data support the conclusions?

Reviewer #1: Yes

Reviewer #2: Yes

Reviewer #3: Yes

2. Has the statistical analysis been performed appropriately and rigorously? 

Reviewer #1: Yes

Reviewer #2: N/A

Reviewer #3: N/A

3. Have the authors made all data underlying the findings in their manuscript fully available?

Reviewer #1: Yes

Reviewer #2: No

Reviewer #3: Yes

4. Is the manuscript presented in an intelligible fashion and written in standard English?

Reviewer #1: Yes

Reviewer #2: Yes

Reviewer #3: Yes

5. Review Comments to the Author

Reviewer #1: The study aims to examine and describe the developmental stages of a an endoparasite copepod Ive ptychoderae, lining inside the acorn worm. Along with the identification of embryonic and post-embryonic development stages, the authors compare the morphological characters in the naupliar stages in order to assess the phylogenetic position of the copepod. Even though more thorough comparative studies are needed to get a more definitive answer to the phylogenetic status of Ive- group, the authors did a great job in highlighting the developmental stages of Ive ptychoderae which could prove to be an important tool for identifying the position of Ive-group and relating it to other copepod groups. I have a few suggestions as follows:

In the Abstract (Line no. 32 : difficulties when investigating…parasitic copepod). There is a typo, taxanomy sould be corrected to taxonomy.

In results (Line no. 186-189: However, due to the nuclei and cleavages……1-, 2-, 4-, 8-, 16-cell stage, blastula, gastrula, and limb bud stages), the sentence is not clear as to what the authors are trying to say I think “due to” needs to be removed If the nuclei and cleavages of eggs were clearly visible.

Line no. 189-191: The duration of embryogenesis spent about 70h……development time(37 h) (Fig 2). could be simplified for readability as “The total duration of embryogenesis was 70h from 1 cell stage to limb bud stage where limb bud stage accounted for more than half of the development time (37h) (Fig 2).

Line no. 356 (The differences between CI and CII were only occurred…….armature of antennule and antenna). “Were” is wrongly placed in the sentence.

Line no. 358 (The duration from CII to CIII was long as well as the duration from CIII to CIV), the sentence is unclear. It can be written as follows: The durations of CII to CIII and CIII to CIV were comparatively longer (than CI to CII?)

In Discussion (Line 413: The copepods…stages) Typo, copepod instead of copepods.

Line 420 (and do not feed during….(2) its naupliar stage may less than six) typo, missing “be” before less)

Could Ive- group be considered as a connector of more primitive cyclopoid group and the recent ones or a primitive form of more recent group since many morphological characters are unique to Ive- (such as 2-segmented endopod in nauplius stage) and its 18S rDNA sequence of Ive- is also quite different.

I also feel that the simplification of body plan should be addressed a little more (Line 406), maybe comparison with other parasitic copepods of the same group (Ive balanoglossi) or other groups can be used as reference to explain the simplification.

Reviewer #2: Reviewer’s comment

The manuscript is well written and intended to provide taxonomic information on the rare known parasitic species of copepods, which is highly ambiguous in terms of taxonomic position and systematics. The parasitic invasion in commercially important species is the major concern as it may lead to large scale mortality leading to economic and ecological repercussions. Therefore, knowing the biology, Lifecyle and phylogeny of parasitic species is of utmost important. I would specially like to mention that, this research has adopted an embryology and developmental approach to address this gap. Laboratory based experiment used in this study is highly imperative, and based on this observation the authors were able to make final call on the taxonomic affinity and different pattern of developmental larval stages.

I believe, this manuscript has a potential of publication in PLOSE ONE. However, few of minor corrections from my side have been highlighted, corrected or deleted in the PDF file and request the authors to kindly incorporate the same as long as those are empirical. Since I am not a native of English speaking but still, I gave try to correct few.

In addition to this, few of my comments are given below.

Comment 1. One of my concerns is that, the authors have only given very little information about the parasitic species and taxonomic and morphological characteristics alone.

An addition of 4-5 sentences in the introduction part regarding the ecosystem role the copepod plays, the ecological effects on the entire ecosystem the parasitic copepods may cause and what possible impact it would bring on the function of the ecosystem? Will do justice for the objective set and addressed.

Comment 2. The authors have done re-infection study, Line 161. It would be highly appreciated, if the authors put stress on why the re-infection study was important what purpose does it fulfill.

Comment 3. Few of the paragraph heading contain only few sentences which either can be merged into related heading if possible. For example, in the 340 under adult header, few more sentences can be added let be very brief though as already elaborated in other document as mentioned by the authors and commented further by the reviewer in the PDF.

Comment 4. At last, why the authors failed to put a separate heading of CONCLUSION though a good conclusive remark has been given into discussion section, last paragraph. A separate conclusive remark with finding will be much easier for the general audience.

Reviewer #3: This study did contribute in understanding the systematics of parasitic copepods. I appreciate the effort of the authors. However, I would like to make few suggestions.

1. Mention a specific number of naupli (Line no: 154) and infected acorn worms sacrificed (Line no: 164) instead of "over thousands" and "one to three".

2. The phrase "approximately 3 d" (Line no: 154) is unclear. Author should clarify or use approprite word insted.

3. Line no: 171, Author needs to provide the purpose of using 85% lactic acid.

4. Number of specimens used for morphological studies of each development stages should be clearly mentioned in material and methods section.

5. Line No: 210, Maintain uniform use of uppercase and lowercase characters.

6. Study has provided good images of specimens. However, proper labelling (eg. Leg numbers mentioned in manuscript, not present in figures) in Figs 4-10 is required.

7. Author has mentioned " eggs developed into next step" (Line no: 146), is it an embryonic development stage or a step involved in methods, needs clarification.

Further, i would like to recommend referring following research articles which will significantly improve this manuscript:

Kumar and rao (1998), doi:10.1093/plankt/20.2.271 and Kumar (2003), doi:10.1127/0003-9136/2003/0157-0351.

6. PLOS authors have the option to publish the peer review history of their article (what does this mean?). If published, this will include your full peer review and any attached files.

Reviewer #1: **Yes: **Malayaj Rai

Reviewer #2: **Yes: **Jawed Equbal

Reviewer #3: **Yes: **Devesh Kumar Yadav

---

## [Author Response · Author response to Decision Letter 0]

4 Jan 2023

Point-by-point responses to the comments 

Comments from Editor: The developmental stages of this family, Iviidae, and the life cycle of I. ptychoderae based on the morphological characters of larvae will fill the knowledge gap and help in the establishment of phylogenetic affiliation. The study is well designed and systematically done. These results would help to resolve the problematic phylogenetic position of the Ive-group, and showcase the approach of utilizing developmental characters (i.e., morphology of immature stages). 

Response: We thank editor for the positive comments and encouragement.

A sentence in the introduction section regarding the role of parasitic copepods on ecosystem structuring need to be explained. Justify the re-infection study in this Ms. 

Response: Following this suggestion, we added a few sentences about the role of parasitic copepods on marine ecosystem in the introduction section of the revised manuscript (line 59-63 in the file “Revised Manuscript with Track Changes”).

Besides the Manuscript will benefit from following published information: doi:10.1093/plankt/20.2.271 and Kumar (2003), doi:10.1127/0003-9136/2003/0157-0351. I am personally curious to learn the post embryonic developmental rate as a function of (i) host availability (ii) temperature and salinity. Overall the manuscript is well written, methods are easily reproducible.

Response: Following this suggestion, we added these two papers (along with another reference) into the Discussion section in our revised manuscript (references 48-50), to point out the limitation of our current study and propose a possible future direction to address this issue (line 528-534). 

Comments from Reviewer #1: The study aims to examine and describe the developmental stages of an endoparasite copepod Ive ptychoderae, living inside the acorn worm. Along with the identification of embryonic and post-embryonic development stages, the authors compare the morphological characters in the naupliar stages in order to assess the phylogenetic position of the copepod. Even though more thorough comparative studies are needed to get a more definitive answer to the phylogenetic status of Ive- group, the authors did a great job in highlighting the developmental stages of Ive ptychoderae which could prove to be an important tool for identifying the position of Ive-group and relating it to other copepod groups. 

Response: Thank you very much for your kind comments.

In the Abstract (Line no. 32 : difficulties when investigating…parasitic copepod). There is a typo, taxanomy should be corrected to taxonomy.

Response: Fixed.

In results (Line no. 186-189: However, due to the nuclei and cleavages……1-, 2-, 4-, 8-, 16-cell stage, blastula, gastrula, and limb bud stages), the sentence is not clear as to what the authors are trying to say I think “due to” needs to be removed If the nuclei and cleavages of eggs were clearly visible.

Response: Thanks for pointing out this problem. We have removed “due to” from this sentence (now line 265).

Line no. 189-191: The duration of embryogenesis spent about 70h……development time (37 h) (Fig 2). could be simplified for readability as “The total duration of embryogenesis was 70h from 1 cell stage to limb bud stage where limb bud stage accounted for more than half of the development time (37h) (Fig 2).

Response: We have followed the suggestion to modify this sentence (line 268-270). 

Line no. 356 (The differences between CI and CII were only occurred…….armature of antennule and antenna). “Were” is wrongly placed in the sentence.

Response: Thanks for pointing out this mistake, and we deleted “were” from this sentence in the revised manuscript (line 451). 

Line no. 358 (The duration from CII to CIII was long as well as the duration from CIII to CIV), the sentence is unclear. It can be written as follows: The durations of CII to CIII and CIII to CIV were comparatively longer (than CI to CII?)

Response: We have followed the suggestion to modify this sentence in the revised manuscript (line 454-455).

In Discussion (Line 413: The copepods…stages) Typo, copepod instead of copepods.

Response: Fixed.

Line 420 (and do not feed during….(2) its naupliar stage may less than six) typo, missing “be” before less)

Response: Fixed. 

Could Ive- group be considered as a connector of more primitive cyclopoid group and the recent ones or a primitive form of more recent group since many morphological characters are unique to Ive- (such as 2-segmented endopod in nauplius stage) and its 18S rDNA sequence of Ive- is also quite different.

Response: We agree with the reviewer that this can be one possible interpretation for our current results. However, as we stated in the Conclusion section, we feel that more evidence with a much broader taxonomic sampling will be needed for further morphological and molecular investigation to support this idea.

I also feel that the simplification of body plan should be addressed a little more (Line 406), maybe comparison with other parasitic copepods of the same group (Ive balanoglossi) or other groups can be used as reference to explain the simplification.

Response: Thanks for your suggestion. Many endo-parasitic copepods as well as Ive balanoglossi have simplification of body plan. Herein, we have added more text in the revised manuscript to describe its transformed body plan and reduced appendages in the RESULTS section (line 432-436) (as the suggestion from reviewer#2). 

Comments from Reviewer #2: The manuscript is well written and intended to provide taxonomic information on the rare known parasitic species of copepods, which is highly ambiguous in terms of taxonomic position and systematics. The parasitic invasion in commercially important species is the major concern as it may lead to large scale mortality leading to economic and ecological repercussions. Therefore, knowing the biology, Lifecycle and phylogeny of parasitic species is of utmost important. I would specially like to mention that, this research has adopted an embryology and developmental approach to address this gap. Laboratory based experiment used in this study is highly imperative, and based on this observation the authors were able to make final call on the taxonomic affinity and different pattern of developmental larval stages.

I believe, this manuscript has a potential of publication in PLOSE ONE. However, few of minor corrections from my side have been highlighted, corrected or deleted in the PDF file and request the authors to kindly incorporate the same as long as those are empirical. Since I am not a native of English speaking but still, I gave try to correct few.

Response: We are very grateful to the reviewer for the helpful comments and the editing suggestions including in the PDF file. We have followed those suggestions to revise this manuscript. 

Comment 1. One of my concerns is that, the authors have only given very little information about the parasitic species and taxonomic and morphological characteristics alone. An addition of 4-5 sentences in the introduction part regarding the ecosystem role the copepod plays, the ecological effects on the entire ecosystem the parasitic copepods may cause and what possible impact it would bring on the function of the ecosystem? Will do justice for the objective set and addressed.

Response: We thank the reviewer for this comment. As stated in our response to Reviewer# 1, we added a few sentences regarding the potential impacts of parasitic copepods on the marine ecosystem in the revised manuscript (line 59-63).

The authors have done re-infection study, Line 161. It would be highly appreciated, if the authors put stress on why the re-infection study was important what purpose does it fulfill.

Response: Thanks for your suggestion. We have added a sentence (line 214-216) to emphasize the important and purpose of re-infection study.

Comment 3. Few of the paragraph heading contain only few sentences which either can be merged into related heading if possible. For example, in the 340 under adult header, few more sentences can be added let be very brief though as already elaborated in other document as mentioned by the authors and commented further by the reviewer in the PDF.

Response: Thanks for your suggestion. In the revised manuscript, those paragraph headings are remerged into four parts (Eggs and the duration of embryonic developments, Morphology of nauplius stages, copepodid stages, and adult stage, and The duration of post-embryonic developments) (lines 258, 281, and 440). In addition, we have added more text to describe the adult stage of I. ptychoderae (line 432-436).

Comment 4. At last, why the authors failed to put a separate heading of CONCLUSION though a good conclusive remark has been given into discussion section, last paragraph. A separate conclusive remark with finding will be much easier for the general audience.

Response: We appreciate this great suggestion! In the revised manuscript, we have added a Conclusion section, summarizing our major findings, limitations, and future research perspectives.

Comments from Reviewer #3: This study did contribute in understanding the systematics of parasitic copepods. I appreciate the effort of the authors. However, I would like to make few suggestions.

1. Mention a specific number of naupli (Line no: 154) and infected acorn worms sacrificed (Line no: 164) instead of "over thousands" and "one to three".

Response: We thank the reviewer for this suggestion. The information about the number of naupli and sacrificed acorn worms are added in the Materials and Methods section in the revised manuscript (line 206-207, and 216).

2. The phrase "approximately 3 d" (Line no: 154) is unclear. Author should clarify or use appropriate word instead.

Response: Thanks for your suggestion. We have modified the sentence to be more specific on the time period (line 207). 

3. Line no: 171, Author needs to provide the purpose of using 85% lactic acid.

Response: Using lactic acid is to soften the carapace of copepod, which makes the observation and dissection process more easily. This information has been added in the revised manuscript (line 227).

4. Number of specimens used for morphological studies of each development stages should be clearly mentioned in material and methods section.

Response: These information has been provided in the text of the Results section. Every stage includes 10 specimens used for morphological studies. For example, we examined 10 specimens of copepodid I (line 347) and II (line 377) stages.

5. Line No: 210, Maintain uniform use of uppercase and lowercase characters.

Response: Thanks for pointing out this problem. It has been fixed (line 296).

6. Study has provided good images of specimens. However, proper labelling (eg. Leg numbers mentioned in manuscript, not present in figures) in Figs 4-10 is required.

Response: Thanks for your suggestion. We have added additional labels in the revised Figs 4-9, to help the readers to identify different appendages more easily. Because Figure 10 is a schematic summary the post-embryonic development of I. ptychoderae, we feel that further labeling on the appendages may distract the readers, and thus we leave it unchanged. We hope this is agreeable to the reviewer. 

7. Author has mentioned " eggs developed into next step" (Line no: 146), is it an embryonic development stage or a step involved in methods, needs clarification.

Response: Sorry about this confusion. It means the next stage of embryonic development. We rewrite this sentence to make it clear (line 197).

Further, i would like to recommend referring following research articles which will significantly improve this manuscript:

Kumar and rao (1998), doi:10.1093/plankt/20.2.271 and Kumar (2003), doi:10.1127/0003-9136/2003/0157-0351.

Response: Many thanks for providing those valuable papers. We have cited these two papers in the revised manuscript (line 528-530).

---

## [Decision Letter · Decision Letter 1]

13 Jan 2023

Embryonic and post-embryonic development in the parasitic copepod Iveptychoderae (Copepoda: Iviidae): insights into its phylogenetic position

PONE-D-22-29055R1

Dear Dr. Jr-Kai Yu

We’re pleased to inform you that your manuscript has been judged scientifically suitable for publication and will be formally accepted for publication once it meets all outstanding technical requirements.

Kind regards,

Ram Kumar, Ph.D.

Academic Editor

PLOS ONE

Additional Editor Comments (optional):

Reviewers' comments:

Reviewer's Responses to Questions

**Comments to the Author**

1. If the authors have adequately addressed your comments raised in a previous round of review and you feel that this manuscript is now acceptable for publication, you may indicate that here to bypass the “Comments to the Author” section, enter your conflict of interest statement in the “Confidential to Editor” section, and submit your "Accept" recommendation.

Reviewer #1: All comments have been addressed

Reviewer #3: All comments have been addressed

2. Is the manuscript technically sound, and do the data support the conclusions?

Reviewer #1: Yes

Reviewer #3: Yes

3. Has the statistical analysis been performed appropriately and rigorously? 

Reviewer #1: Yes

Reviewer #3: N/A

4. Have the authors made all data underlying the findings in their manuscript fully available?

Reviewer #1: Yes

Reviewer #3: Yes

5. Is the manuscript presented in an intelligible fashion and written in standard English?

Reviewer #1: Yes

Reviewer #3: Yes

6. Review Comments to the Author

Reviewer #1: All the suggestions and comments have been addressed to by the authors. I have no additional comments or concerns regarding the manuscript.

Reviewer #3: (No Response)

7. PLOS authors have the option to publish the peer review history of their article (what does this mean?). If published, this will include your full peer review and any attached files.

Reviewer #1: **Yes: **Malayaj Rai

Reviewer #3: **Yes: **Devesh Kumar Yadav

---

## [Editor Report · Acceptance letter]

27 Feb 2023

PONE-D-22-29055R1 

Embryonic and post-embryonic development in the parasitic copepod *Ive ptychoderae* (Copepoda: Iviidae): insights into its phylogenetic position 

Dear Dr. Yu:

I'm pleased to inform you that your manuscript has been deemed suitable for publication in PLOS ONE. Congratulations! Your manuscript is now with our production department. 

Kind regards, 

on behalf of

Professor Ram Kumar 

Academic Editor

PLOS ONE